# Association between methylmalonic acid and Alpha-Klotho in American adults: A cross-sectional study

Ronghui Bao[1,2‡]*, Hongyan Qi[1,2], Lei Liao[1,2], Qinqin Yu[1,2], Jie Li[1,2], Rong Liu[1,2], Chang Zhou[1,2]

1 The First College of Clinical Medical Science, China Three Gorges University, Yichang, China,
2 Yichang Central People's Hospital, Yichang, China

‡ This author first author on this work.
* baoronghui0128@ctgu.edu.cn

## Abstract

### Objectives

Methylmalonic acid is a surrogate biomarker of mitochondrial dysfunction and oxidative stress. Serum soluble α-Klotho, as a key anti-aging factor, is regarded as one of the biomarkers of aging. The correlation between Methylmalonic Acid (MMA) and Alpha-Klotho (α-Klotho) remains uncertain. This study aims to explore the relationship between MMA and Alpha-Klothoα-Klotho in American adults.

### Materials and methods

Based on the availability of complete biochemical assays for MMA and α-Klotho, we restricted our analyses to the 2011–2014 cycles of the National Health and Nutrition Examination Survey (NHANES), which included 5,216 eligible participants with comprehensive laboratory data. Due to the limited amount of data, there may be a selection bias. In this study, MMA serves as the independent variable while α-Klotho functions as the dependent variable. MMA is a categorical variable, while α-Klotho is a categorical variable. The covariates examined include sociodemographic factors, lifestyle choices, and various systemic diseases. Logistic regression was used to assess the associations between covariates and different independent groups. To explore the relationship between serum MMA levels and α-Klotho, we employed three models: Model 1 adjusted for age, gender, race/ethnicity, education, Marital status, Poverty Income Ratio (PIR), smoking, Body Mass Index (BMI) and Physical activity (PA). Model 2 included all variables from Model 1 plus Cardiovascular diseases (CVD), Hypertension, Diabetes Mellitus (DM). Considering the collinearity problem, Model 3 adjusted for Marital status, PIR, Educations, Smoke, PA, CVD,

**Data availability statement:** These survey data are free and publicly available, and can be downloaded directly from the NHANES website (https://www.cdc.gov/nchs/nhanes/) by users and researchers worldwide.

**Funding:** Medical Health Research Project of Science and Technology Bureau of Yichang City, Hubei Province (A23-1-021).

**Competing interests:** NO authors have competing interests.

Hypertension, DM, Glycated Hemoglobin A1c (HbA1c), High-Density Lipoprotein (HDL), Vitamin B12 (VitB12), **Estimated** Glomerular Filtration Rate (eGFR).

## Results

A total of 5216 participants were included in the study. Among them, 29.4% (1531 participants) had α-Klotho levels below 704.00 pg/mL, while 70.6% (3685 participants) had levels of 704 pg/mL or greater. Compared to individuals with low serum MMA levels Q1 (<120 nmol/L), the adjusted odds ratios (OR) for α-Klotho with MMA levels in Q2 (120–175 nmol/L), Q3 (175–250 nmol/L), and Q4 (≥250 nmol/L) were 0.81 (95% CI: 0.72 to 0.91, p=0.001), 0.81 (95% CI: 0.71 to 0.93, p=0.003), and 0.80 (95% CI: 0.68 to 0.94, p=0.007), respectively.The correlation between MMA levels and α-Klotho was linearly correlated (non-linearity: P=0.62).

## Conclusion

Our findings indicate a significant association between higher MMA levels and lower serum α-Klotho concentrations, It suggests that mitochondrial dysfunction may play a significant role in the aging process. Further research is necessary to validate these findings. This relationship warrants further investigation to clarify its implications for aging and health outcomes.

## Introduction

Mitochondria-derived methylmalonic acid is a surrogate biomarker of mitochondrial dysfunction and oxidative stress. Methylmalonic acid (MMA) is a byproduct of the metabolic pathway of propionic acid [1], Elevated MMA concentrations can disrupt mitochondrial homeostasis, the metabolism of MMA is dependent on the optimal operation of mitochondria. A growing compilation of research indicates that MMA plays a significant role in the development of mitochondrial dysfunction and oxidative stress in both in vitro and in vivo studies [2,3]. Numerous investigations have established that increased circulating levels of MMA are reflective of the aging process.

The KL gene is responsible for encoding the α-Klotho protein [4], which is a pivotal anti-aging gene. This protein can be found in both membrane-bound and soluble forms [5], and it uniquely governs essential metabolic processes that are crucial for health and disease [6]. α-Klotho is predominantly expressed in the kidneys and is vital for the regulation of phosphate and calcium homeostasis,thereby influencing aging and the onset of chronic diseases [7,8]. Although there is mounting evidence regarding the roles of α-Klotho and methylmalonic acid in the aging process,our understanding of their interaction and clinical significance as biomarkers remains insufficient. Limited studies have systematically investigated the association between α-Klotho levels and methylmalonic acid levels [9]. Additionally, the potential of these biomarkers to predict health outcomes in older adults has not been extensively studied.

The primary aim of this research was to evaluate the relationship between serum α-Klotho levels, MMA levels, and various health-related factors. We utilized data from the National Health and Nutrition Examination Survey (NHANES) within a cross-sectional framework to investigate their possible functions as biomarkers associated with aging and metabolic dysfunction.

## Materials and methods

This cross-sectional study utilized NHANES data collected by the Centers for Disease Control and Prevention between 2011 and 2014. The purpose of the NHANES survey is to assess the health and nutritional status of non-hospitalized Americans using stratified multistage probabilistic methods [10]. NHANES collects demographic and detailed health information.This is done through home visits, screenings, and laboratory tests carried out by mobile screening centers (MECs). NHANES is authorized by the National Center for Health Statistics (NCHS) Ethics Review Committee, and all participants provide written informed consent before taking part [11]. Data from NHANES can be accessed on the NHANES website (https://wwwn.cdc.gov/nchs/nhanes/Default.aspx), as of September 1, 2024. In our study, participants aged 40 years and above who had complete data on serum α-Klotho, methylmalonic acid, and covariates were included. We excluded pregnant women and participants with missing α-Klotho, MMA data. To ensure participant confidentiality, the data shared publicly have been rigorously anonymized, with all personal identifying information permanently removed in accordance with journal policy and ethical guidelines.

### Measurement of methylmalonic acid levels

Blood samples were obtained through venepuncture at mobile examination centers (MEC) in accordance with established protocols [12]. The measurement of methylmalonic acid (MMA) was conducted in plasma and/or serum, with a preference for plasma. Prior investigations have indicated that the concentrations of MMA in serum and plasma exhibit identical reference ranges and comparable coefficients of variation (CV). Detailed detection procedures can be found on the NHANES website. Given that both isomers possess the same molecular weight,it has been established that chromatographic techniques are effective in separating MMA from succinic acid. The quantification of MMA was performed utilizing gas chromatography coupled with liquid chromatography-mass spectrometry. The quantification of MMA was achieved by comparing the peak areas of MMA with those of the isotope-labeled d3MMA.A favorable linear correlation for MMA was observed within the concentration range of 50–2000 nmol/L. The total coefficient of variation (CV) ranged from 4% to 10%, and the average recovery rate was found to be 96.0%±1.9%,as specified in the NHANES laboratory protocols. Serum MMA was extracted and analyzed using the LC-MS/MS method, while the contract laboratory performs random checks on 2% of all samples.

### Measurement of serum α-klotho levels

Serum α-klotho levels were analyzed in frozen samples from individuals aged 40–79 years,which were collected during NHANES from 2011 to 2014 over the period from 2019 to 2020. Fresh-frozen serum samples, stored at −80°C, were sent from the Centers for Disease Control and Prevention to the Northwest Lipid Metabolism and Diabetes Research Laboratories at the University of Washington's Division of Metabolism, Endocrinology, and Nutrition. Serum α-klotho levels were measured with an ELISA kit from IBL International in Japan. For quality assurance, we used the average of two duplicate analyses. Samples with duplicate results over 10% were reanalyzed. The assay sensitivity measured was 4.33 pg/mL. To assess assay linearity,we used two samples with very high and high klotho concentrations at various dilutions.2011–2014 α-klotho levels ranging from 151.3 to 3456 pg/mL, with a mean of 698.0 pg/mL [13]. The final values of all samples exceeded this limit, so no interpolation was performed. All specimen optical densities were above the 50 pg mL$^{-1}$ lower limit of quantification (LLOQ) established for the kit; therefore no sample required extrapolation below the lowest standard [14].

## Covariates

Various potential covariates were assessed based on the literature [15–17]. These include age, gender, marital status, race/ethnicity, education level, Poverty Income Ratio(PIR), smoking status, physical activity(PA), Body Mass Index (BMI), Glycosylated Hemoglobin (HDL), High-Density Lipoprotein (HbA1c), **Estimated** Glomerular Filtration Rate (eGFR), Vitamin B12 (VitB12), Cardiovascular diseases (CVD), Hypertension, Diabetes Mellitus (DM). Age groups were classified as 40–65 years or 65 years and older. Sex was categorized as male or female, Race/ethnicity included non-Hispanic white,non-Hispanic black, Mexican American, and other races (including multiracial and other Hispanic). BMI was defined as $< 18.5$ kg/m$^2$, Normal weight:18.5–24.9 kg/m$^2$, Overweight: 25.0–29.9 kg/m$^2$,Obesity: BMI $\geq 30.0$ kg/m$^2$; Marital status was classified as married, living with a partner, or living alone. Education levels were classified into three categories: less than 9 years, 9–12 years, and more than 12 years. Household income was categorized into three groups based on the poverty income ratio (PIR) as reported by the U.S government: low (PIR $\leq 1.3$), medium (PIR $> 1.3$ to 3.5),and high (PIR $> 3.5$). Smoking status was classified based on previous literature into three categories:never smokers(those who have smoked fewer than 100 cigarettes), current smokers,and former smokers (those who quit after smoking more than 100 cigarettes), eGFR ($\geq 60$ and 30–60 mL/min/1.73m$^2$), Vitamin B12 $< 400$ and $\geq 400$ pmol/L).

## Statistical analysis

Because the sample size relied solely on the available data, no a priori calculation of statistical power was conducted. Consequently, analyses were performed using R Software (version 4.2.1;R Foundation for Statistical Computing;http://www.Rproject.org), the R Survey package (version 4.1−1), and Free Statistics Software (version 1.9.2; Beijing Free Clinical Medical Technology Co.Ltd). In all analyses, a two-sided p-value< 0.05 was taken to indicate statistical significance.

## Results

### Study population

A total of 11,329 participants aged 40 years or older completed the interview. We excluded participants for several reasons: pregnant women (n = 122); missing methylmalonic acid data (n = 1,125); and missing α-Klotho data (n = 4,866). Ultimately, this cross-sectional study included 5,216 participants from the NHANES dataset collected between 2011 and 2014. Fig 1 illustrates the detailed inclusion and exclusion process.

### Baseline Characteristics

Table 1 Presents the baseline characteristics of all subjects categorized by serum methylmalonic acid (MMA) quartiles. Among the subjects, 1531 (29.4%) had α-Klotho levels below 704.00 pg/mL, while 3685 (70.6%) had levels at or above this threshold. The average age of the study participants was 57.4 years (SD = 10.7), and 72.1% were men. Individuals with high MMA levels tended to be 40-65 y, predominantly male, often married or living with a partner, and mostly non-Hispanic white. They also had a higher body mass index, lower household income, and engaged in less physical activity. Furthermore, the rates of hypertension were higher among these individuals.

### Relationship between serum MMA and α-Klotho

When serum levels of methylmalonic acid were analyzed using quartiles, there was a negative association between serum levels of MMA and α-Klotho (after adjusting for potential confounders).Compared with individuals with low serum MMA levels in Q1 (MMA< 120 nmol/L),the adjusted OR values for MMA levels and α-Klotho in Q2 (120–175 nmol/L), Q3 (175–250 nmol/L),and Q4 ($\geq 250$ nmol/L) were 0.89 (95% CI: 0.83~0.95, p = 0.001), 0.93 (95% CI: 0.86~1.01, p = 0.079),and 0.88 (95% CI: 0.8~0.96, p = 0.006 (Table 2), respectively. The analysis results after gradually adjusting for confounding factors showed that there was a linear correlation between the MMA level and α-Klotho.

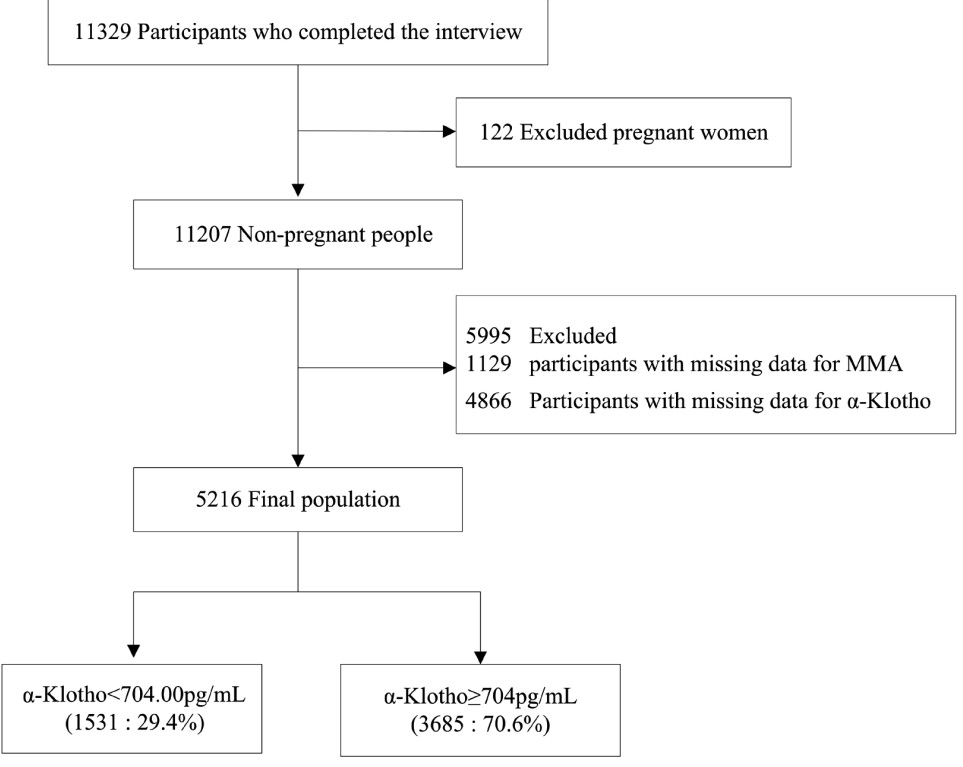

**Fig 1. Flow diagram of participant inclusion and exclusion of NHANES 2011–2014.**

Fig 2 The odds ratio of serum alpha-Klotho, The solid line indicates the predicted value, while the dashed line shows the 95% confidence interval. The analysis was adjusted for multiple variables, including Marital status, PIR, Education, PA, Smoking, CVD, Hypertension, DM, HbA1c, HDL, eGFR, and VitB12. It is important to note that only 95% of the data is displayed in the figure for clarity. The correlation between serum MMA levels and α-Klotho was linear (P = 0.62) in RCS (Fig 2).

## Stratified analyses based on additional variables

The relationship between MMA and α-klotho is divided into several subgroups by using additional variables for hierarchical analysis. After adjustment for multiple comparisons, no additive interactions were observed for sex, age, smoking, VitB12, eGFR, CVD, hypertension, or diabetes across these grouping variables. This indicates that the MMA–α-Klotho association is independent of vitamin B12 and eGFR levels, and the findings remained robust in the stratified analyses (Fig 3).

## Discussion

After adjusting for possible confounding factors such as glomerular filtration rate and VitB12 in this study, it was found that the serum MMA level was negatively correlated with α-Klotho (S1 Fig). Our findings show a significant association between elevated MMA levels and reduced serum α-Klotho concentrations.This might suggest highlighting the potential role of mitochondrial dysfunction in the aging process. Compared with the weighted analysis, the effect estimates and P-values of the results changed slightly, but the overall trend remained consistent.

**Table 1. Population characteristics classified by MMA.**

| Variables | Total (n=5216) | Q1 (n=1564) | Q2 (n=1897) | Q3 (n=1058) | Q4 (n=697) | p |
|---|---|---|---|---|---|---|
| **Age, Mean±SD** | 57.4±10.7 | 54.1±10.1 | 57.3±10.5 | 59.5±10.7 | 61.6±10.4 | < 0.001 |
| **Age(y), n (%)** | | | | | | < 0.001 |
| 40-65 | 3759 (72.1) | 1274 (81.5) | 1394 (73.5) | 688 (65) | 403 (57.8) | |
| ≥65 | 1457 (27.9) | 290 (18.5) | 503 (26.5) | 370 (35) | 294 (42.2) | |
| **Sex, n (%)** | | | | | | 0.002 |
| Male | 2484 (47.6) | 689 (44.1) | 906 (47.8) | 535 (50.6) | 354 (50.8) | |
| female | 2732 (52.4) | 875 (55.9) | 991 (52.2) | 523 (49.4) | 343 (49.2) | |
| **BMI_kg.m², Mean±SD** | 29.7±6.9 | 29.5±6.7 | 29.4±6.5 | 30.0±7.3 | 30.0±7.6 | 0.1 |
| **BMI(kg.m²), n (%)** | | | | | | 0.112 |
| 18.5 | 51 (1.0) | 14 (0.9) | 15 (0.8) | 7 (0.7) | 15 (2.2) | |
| 18.5-24.9 | 1252 (24.3) | 375 (24.2) | 461 (24.5) | 247 (23.7) | 169 (24.6) | |
| 25-30 | 1761 (34.1) | 540 (34.9) | 650 (34.6) | 352 (33.7) | 219 (31.9) | |
| ≥30 | 2093 (40.6) | 618 (39.9) | 754 (40.1) | 438 (42) | 283 (41.3) | |
| **Race, n (%)** | | | | | | < 0.001 |
| Non-Hispanic White | 2116 (40.6) | 396 (25.3) | 821 (43.3) | 547 (51.7) | 352 (50.5) | |
| Non-Hispanic Black | 1162 (22.3) | 440 (28.1) | 414 (21.8) | 196 (18.5) | 112 (16.1) | |
| Mexican American | 642 (12.3) | 272 (17.4) | 203 (10.7) | 89 (8.4) | 78 (11.2) | |
| Other Hispanic | 539 (10.3) | 168 (10.7) | 202 (10.6) | 102 (9.6) | 67 (9.6) | |
| Other Race | 757 (14.5) | 288 (18.4) | 257 (13.5) | 124 (11.7) | 88 (12.6) | |
| **Marital status, n (%)** | | | | | | < 0.001 |
| Married | 3076 (59.0) | 974 (62.3) | 1131 (59.6) | 618 (58.5) | 353 (50.6) | |
| Never married | 494 (9.5) | 151 (9.7) | 161 (8.5) | 106 (10) | 76 (10.9) | |
| Living with partner | 230 (4.4) | 79 (5.1) | 87 (4.6) | 40 (3.8) | 24 (3.4) | |
| Other | 1415 (27.1) | 360 (23) | 518 (27.3) | 293 (27.7) | 244 (35) | |
| **PIR, n (%)** | | | | | | < 0.001 |
| ≤1.30 | 1525 (31.6) | 405 (28.3) | 521 (29.4) | 320 (33.1) | 279 (42.8) | |
| 1.31-3.50 | 1626 (33.7) | 473 (33.1) | 594 (33.5) | 333 (34.5) | 226 (34.7) | |
| >3.50 | 1670 (34.6) | 553 (38.6) | 657 (37.1) | 313 (32.4) | 147 (22.5) | |
| **Education1, n (%)** | | | | | | 0.01 |
| Less than high school | 1285 (24.6) | 380 (24.3) | 430 (22.7) | 274 (25.9) | 201 (28.8) | |
| High school or equivalent | 1111 (21.3) | 308 (19.7) | 426 (22.5) | 223 (21.1) | 154 (22.1) | |
| Above high school | 2819 (54.1) | 876 (56) | 1040 (54.9) | 561 (53) | 342 (49.1) | |
| **Smoker, n (%)** | | | | | | < 0.001 |
| never | 2787 (53.5) | 922 (59) | 994 (52.4) | 527 (49.9) | 344 (49.4) | |
| former | 1462 (28.0) | 380 (24.3) | 556 (29.3) | 317 (30) | 209 (30) | |
| current | 965 (18.5) | 261 (16.7) | 347 (18.3) | 213 (20.2) | 144 (20.7) | |
| **PA Median (IQR)** | 840.0 (0.0, 2880.0) | 840.0 (0.0, 2880.0) | 840.0 (0.0, 2880.0) | 720.0 (0.0, 2640.0) | 700.0 (0.0, 2640.0) | 0.013 |
| **HbA1c n (%)** | | | | | | < 0.001 |
| <6.5% | 4374 (84.0) | 1295 (82.9) | 1632 (86.1) | 898 (85) | 549 (79) | |
| ≥6.5% | 836 (16.0) | 268 (17.1) | 263 (13.9) | 159 (15) | 146 (21) | |
| **HDL-C _mg.dl, Mean± SD** | 53.1±16.0 | 54.1±15.1 | 53.4±15.9 | 52.5±17.3 | 51.3±16.5 | < 0.001 |
| **VitB12, Median (IQR)** | 385.2 (282.7, 546.9) | 453.1 (335.1, 633.2) | 400.7 (298.2, 563.1) | 341.7 (256.1, 456.6) | 280.4 (200.0, 402.9) | < 0.001 |
| **VitB12(pmol/L), n (%)** | | | | | | < 0.001 |
| <400 | 2769 (53.1) | 621 (39.8) | 943 (49.7) | 687 (64.9) | 518 (74.4) | |

*(Continued)*

https://doi.org/10.1371/journal.pone.0337285 December 30, 2025

**Table 1.** (Continued)

| Variables | Total (n = 5216) | Q1 (n = 1564) | Q2 (n = 1897) | Q3 (n = 1058) | Q4 (n = 697) | p |
|---|---|---|---|---|---|---|
| ≥ 400 | 2444 (46.9) | 941 (60.2) | 954 (50.3) | 371 (35.1) | 178 (25.6) | |
| **Creatinine_mg.dl, Mean ± SD** | 0.9 ± 0.6 | 0.8 ± 0.2 | 0.9 ± 0.2 | 1.0 ± 0.3 | 1.3 ± 1.4 | < 0.001 |
| **eGFR, Mean ± SD** | 86.2 ± 20.5 | 96.3 ± 16.1 | 87.0 ± 16.9 | 79.2 ± 19.7 | 72.2 ± 26.2 | < 0.001 |
| **eGFR(mL/min), n (%)** | | | | | | < 0.001 |
| < 60 | 533 (10.2) | 30 (1.9) | 118 (6.2) | 189 (17.9) | 196 (28.1) | |
| ≥60 | 4681 (89.8) | 1533 (98.1) | 1779 (93.8) | 868 (82.1) | 501 (71.9) | |
| **CVD, n (%)** | | | | | | < 0.001 |
| no | 4557 (87.4) | 1434 (91.7) | 1686 (88.9) | 884 (83.6) | 553 (79.5) | |
| yes | 658 (12.6) | 130 (8.3) | 211 (11.1) | 174 (16.4) | 143 (20.5) | |
| **Hypertension, n (%)** | | | | | | < 0.001 |
| no | 2416 (46.3) | 838 (53.6) | 886 (46.7) | 452 (42.7) | 240 (34.4) | |
| yes | 2800 (53.7) | 726 (46.4) | 1011 (53.3) | 606 (57.3) | 457 (65.6) | |
| **DM, n (%)** | | | | | | < 0.001 |
| no | 3889 (74.6) | 1164 (74.4) | 1462 (77.1) | 793 (75) | 470 (67.4) | |
| yes | 1327 (25.4) | 400 (25.6) | 435 (22.9) | 265 (25) | 227 (32.6) | |
| **α-Klotho (pg/mL), n (%)** | | | | | | < 0.001 |
| <704.00 | 1531 (29.4) | 403 (25.8) | 560 (29.5) | 328 (31) | 240 (34.4) | |
| ≥704.00 | 3685 (70.6) | 1161 (74.2) | 1337 (70.5) | 730 (69) | 457 (65.6) | |

Q1: MMA < 120nmol/L, Q2: MMA:120–175nmol/L, Q3: MMA:175–250nmol/L, Q4: MMA > 250nmol/L.

**Table 2. Associations between serum MMA and α-Klotho in the multiple regression mode.**

| MMA (nmol/L) | No. | Crude | P | Model 1 | P | Model2 | P | Model3 | P |
|---|---|---|---|---|---|---|---|---|---|
| | | | | **OR,95%CI** | | | | | |
| MMA Quartiles | 26080 | −0.1(−0.12~−0.08) | <0.001 | −0.07 (−0.09~−0.05) | <0.001 | −0.06 (−0.08~−0.04) | <0.001 | −0.05(−0.07~−0.03) | <0.001 |
| Q1<120 | 7820 | 1(Ref) | | 1(Ref) | | 1(Ref) | | 1(Ref) | |
| Q2: 120–175 | 9485 | 0.83 (0.77~0.89) | <0.001 | 0.91 (0.85~0.98) | 0.01 | 0.91 (0.85~0.98) | 0.012 | 0.89 (0.83~0.95) | 0.001 |
| Q3: 175–250 | 5290 | 0.77 (0.72~0.83) | <0.001 | 0.91 (0.84~0.98) | 0.017 | 0.92 (0.84~0.99) | 0.033 | 0.93 (0.86~1.01) | 0.079 |
| Q4 ≥ 250 | 3485 | 0.66 (0.61~0.72) | <0.001 | 0.8 (0.73~0.88) | <0.001 | 0.82 (0.75~0.9) | <0.001 | 0.88 (0.8~0.96) | 0.006 |
| Trend test | 26080 | | <0.001 | | <0.001 | | <0.001 | | 0.013 |

Q, quartiles; OR, odds ratio; CI, confidence interval; Ref: reference.

Model1: Adjusted for Age, Sex, Race, BMI, Marital status, PIR, Education, PA, Smoking.

Model2: Model1 + Hypertension, DM, CVD.

Model3: Model2 + HbA1c, Hdl_cholesterol, VitB12, eGFR.

α-Klotho is a well-known protein linked to anti-aging, while mitochondria play a crucial role in regulating various key aspects of the aging process [18]. The relationship between mitochondrial dysfunction and aging is an important topic in the field of aging research. The direct association analysis of MMA on α-klotho has not been reported in the literature [19]. NHANES provided us with a unique opportunity to assess whether there is an association between MMA and α-klotho, as well as the dose-response relationship between the two, while accounting for various covariates and conducting a series of stratified analyses.

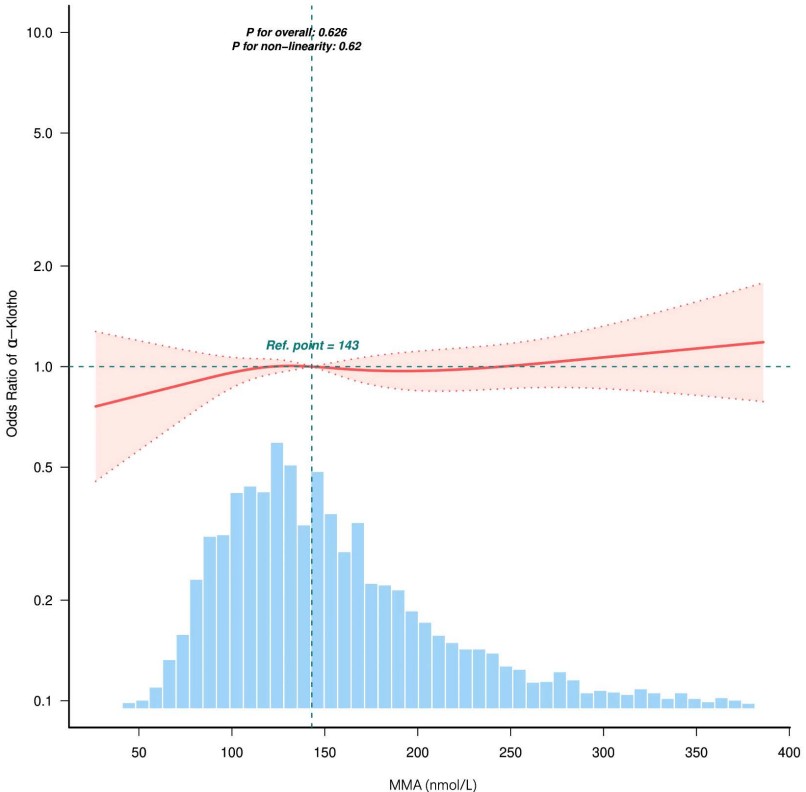

**Fig 2. The relationship between methylmalonic acid levels and the Odds Ratio of serum α-Klotho.**

Details of the underlying mechanism Serum methylmalonic acid (MMA), as a surrogate biomarker of mitochondrial dysfunction, plays a key role in aging and can predict poor prognosis in adults [20]. The buildup of MMA can result in oxidative stress, harm mitochondrial function, disrupt cellular energy metabolism, and ultimately cause cell death [21]. Studies have shown that circulating MMA is linked to accelerated aging phenotypes (PAA) [2], while the clearance of mitochondria in senescent cells reduces the aging-related secretory phenotype (SASP) [22,23]. Additionally,this association remains unaffected by vitamin B12 status and renal function within the general population [24]. This study adjusted for GFR and VitB12, and found that the correlation between MMA and α-Klotho persisted 0.8 (95%CI: 0.68–0.94, p = 0.007).This indicates that the correlation is not influenced by eGFR or VitB12 levels.

The human KL gene encodes the α-Klotho protein, which is recognized as an anti-aging protein [25]. It exists in both membrane-bound and soluble forms,which regulate various metabolic processes essential for health and disease.There are three types of α-Klotho proteins:full-length transmembrane α-Klotho, soluble α-Klotho, and secreted α-Klotho [26]. The full-length α-Klotho protein is a transmembrane protein with two distinct glycohydrolase domains, which are crucial for its activity and function [25]. Klotho deficiency is linked to various age-related diseases, including cancer, high blood pressure, and kidney disease. In humans, serum α-Klotho levels decline with age, particularly after 40y [27]. In mice, KL gene overexpression can prolong lifespan, while KL gene mutation can shorten lifespan. Based on its various known and unknown protective properties,upregulating the Klotho gene may be a possible way to treat or prevent aging-related complications [28].

Our findings are biologically plausible. In vitro studies show that methylmalonyl-CoA accumulation—reflected by elevated serum MMA—inhibits mitochondrial complex IV, increases reactive oxygen species (ROS) and subsequently

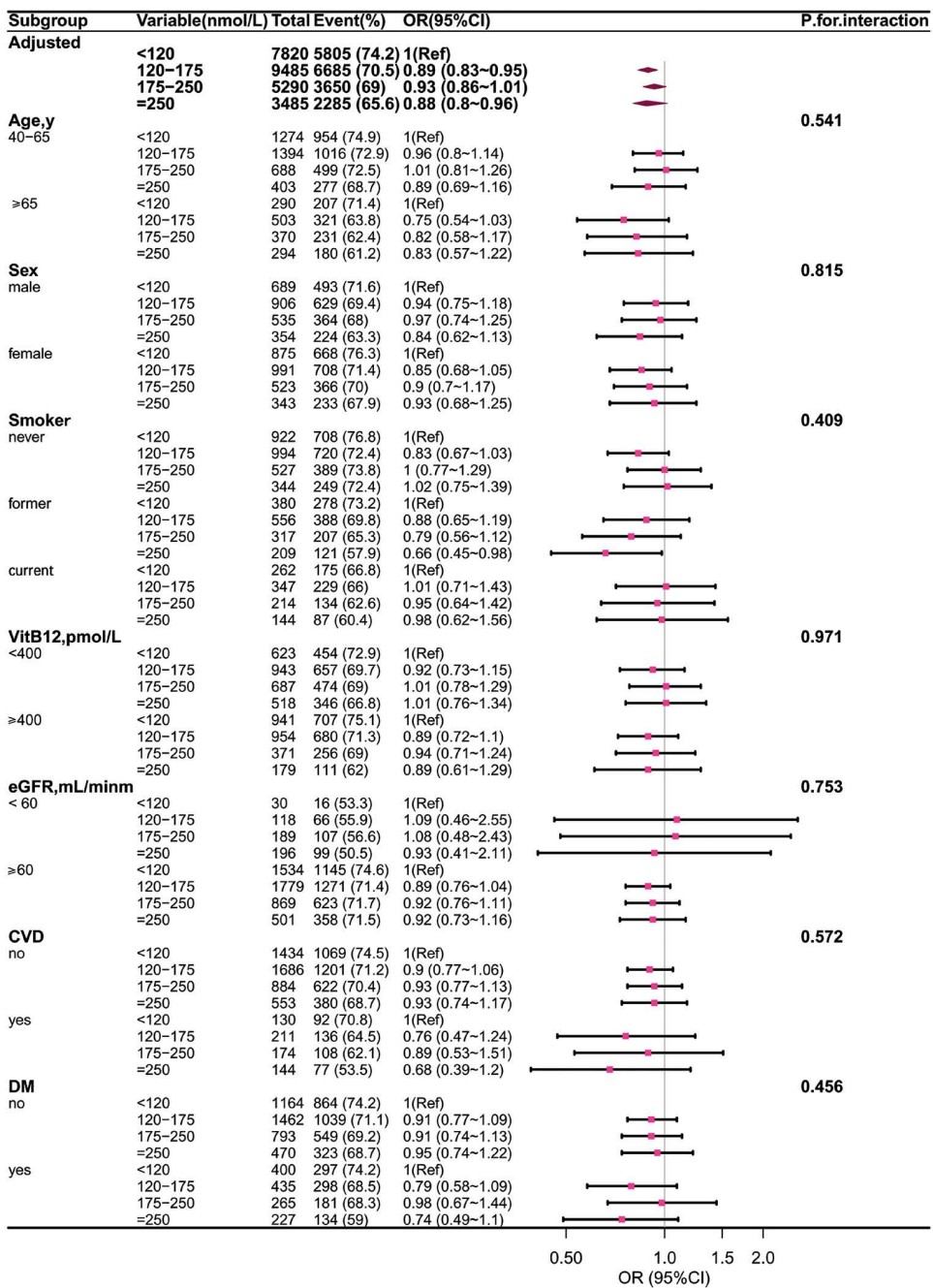

**Fig 3. In this study, the effects of MMA content on α-Klotho in different subgroups were analyzed by forest maps.**

down-regulates klotho expression in renal tubule cells [29]. Conversely, klotho protein preserves mitochondrial integrity by up-regulating SOD2 and PGC-1α, thereby reducing ROS generation [30]. Thus, higher circulating MMA may both indicate and induce mitochondrial oxidative stress, creating a feedback loop that lowers α-Klotho levels; this provides a mechanistic explanation for the inverse association observed in the present cross-sectional analysis.

This study is, to our knowledge, the first to report the link between serum MMA and α-Klotho levels in a representative sample of American adults. Besides its originality, a major strength of this study is the large sample size from NHANES, which represents the broader American population [10]. One advantage is that all data were gathered using standardized interviews, physical examinations, and laboratory tests, which helps minimize measurement bias. However, the study has some limitations. First, the α-Klotho data were collected only between 2011 and 2014, which limited our ability to use NHANES data from other time periods for further verification. Second, even with the use of regression models and stratified analyses to account for potential confounding factors, we cannot completely rule out the residual effects of unmeasured or unknown variables. Third, the findings are based on a survey of American adults and further research is needed to determine if they apply to other populations. Finally, because of the limitations of cross-sectional studies, we cannot establish a causal relationship between MMA and α-klotho; this relationship needs to be confirmed through future longitudinal studies. One limitation of our study is the absence of direct measurements for specific biomarkers that indicate mitochondrial dysfunction or oxidative stress, such as lactate levels [31] and Fibroblast Growth Factor21 (FGF21) [32]. Incorporating these additional biomarkers in future research will significantly improve our understanding of the underlying biological mechanisms.

## Conclusions

This study finds that higher levels of MMA are associated with lower levels of serum α-Klotho. Our research suggests that MMA may influence serum α-Klotho levels either directly or indirectly. Considering that MMA serves as a biomarker for mitochondrial dysfunction, the results indicate that mitochondrial dysfunction might be a contributing factor in the aging processThese cross-sectional associations between serum MMA and α-Klotho warrant confirmation in longitudinal and mechanistic studies before inferring a causal role for mitochondrial dysfunction in aging.

## Supporting information

**S1 Fig. The scatter plot indicates that there is a linear relationship between MMA and α-Klotho.**
(PNG)

**S1 File. Anonymized.**
(CSV)

**S2 File. README.**
(DOCX)

## Author contributions

**Data curation:** Lei Liao.

**Funding acquisition:** Ronghui Bao.

**Methodology:** Jie Li, Rong Liu.

**Resources:** Ronghui Bao.

**Validation:** Chang Zhou.

**Writing – original draft:** Ronghui Bao.

**Writing – review & editing:** Hongyan Qi, Qinqin Yu.

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
