## [Decision Letter · Decision Letter 0]

4 Mar 2025

Dear Dr. ronghui,

Thank you for submitting your manuscript to PLOS ONE. After careful consideration, we feel that it has merit but does not fully meet PLOS ONE’s publication criteria as it currently stands. Therefore, we invite you to submit a revised version of the manuscript that addresses the points raised during the review process.

We look forward to receiving your revised manuscript.

Kind regards,

Nafisa M. Jadavji, PhD, MSc, BSc

Academic Editor

PLOS ONE

- https://doi.org/10.3389/fmed.2023.1269671

- https://doi.org/10.1016/j.scitotenv.2022.156938

In your revision ensure you cite all your sources (including your own works), and quote or rephrase any duplicated text outside the methods section. Further consideration is dependent on these concerns being addressed.

“Medical Health Research Project of Science and Technology Bureau of Yichang City, Hubei Province (A23-1-021).”

5. Please upload a copy of Figure 2, to which you refer in your text on page 4. If the figure is no longer to be included as part of the submission please remove all reference to it within the text.

6. Please include your tables as part of your main manuscript and remove the individual files. Please note that supplementary tables (should remain/ be uploaded) as separate "supporting information" files

Reviewers' comments:

Reviewer's Responses to Questions

**Comments to the Author**

1. Is the manuscript technically sound, and do the data support the conclusions?

Reviewer #1: No

Reviewer #2: Yes

2. Has the statistical analysis been performed appropriately and rigorously?

Reviewer #1: Yes

Reviewer #2: Yes

3. Have the authors made all data underlying the findings in their manuscript fully available?

Reviewer #1: No

Reviewer #2: Yes

4. Is the manuscript presented in an intelligible fashion and written in standard English?

Reviewer #1: No

Reviewer #2: Yes

Reviewer #1: Major Issues for Revision

1. Fundamental Conceptual Flaws and Oversimplifications

The study claims that a negative association between MMA and α-Klotho suggests mitochondrial dysfunction's role in aging. However, correlation does not imply causation, especially in a cross-sectional study. The entire premise that MMA directly reflects mitochondrial dysfunction is questionable.

MMA accumulation is more commonly linked to vitamin B12 deficiency and renal impairment rather than mitochondrial dysfunction.

No direct mechanistic link between MMA and α-Klotho is explored—only speculation.

The authors cite literature suggesting MMA as a mitochondrial dysfunction marker, but many of these studies are context-specific (e.g., genetic disorders, severe metabolic conditions) and cannot be extrapolated to the general population.

→ Solution: The authors need to discuss alternative explanations for their findings (e.g., renal impairment, metabolic dysfunction) rather than assuming mitochondrial dysfunction.

2. Weakness in Methodology and Study Design

Cross-sectional design: This prevents causal inference but is not acknowledged adequately. The manuscript implies a directional effect (MMA → α-Klotho decrease), which is misleading.

Many references in the discussion assume causality without direct mechanistic validation.

Longitudinal or interventional studies are needed to confirm such claims.

Serum MMA measurements:

The study relies entirely on NHANES data but does not provide enough quality control details for MMA measurements.

MMA levels fluctuate due to dietary factors, renal function, and transient metabolic changes, which are not sufficiently controlled.

The authors state that MMA was measured by LC-MS/MS, but no intra- or inter-assay variation data are reported.

α-Klotho measurements:

The ELISA method is reported, but batch effects and assay standardization issues are not discussed.

Why were data from 2011–2014 used exclusively when NHANES has additional datasets? This could lead to selection bias.

The reference range used (285.8–1638.6 pg/mL) is outdated and may not reflect inter-laboratory variations.

→ Solution: Acknowledge cross-sectional study limitations, discuss alternative explanations (e.g., renal dysfunction), and provide detailed quality control measures for biochemical assays.

3. Covariate Adjustments Are Superficial and Inadequate

The study controls for confounders (BMI, HbA1c, renal function, etc.), but does not provide a detailed justification for why these specific variables were chosen.

Renal function (eGFR, creatinine levels) is completely missing as a covariate, which is a major oversight.

MMA is strongly influenced by kidney function, and α-Klotho is also a renal protein.

Without renal function adjustments, the reported association could be entirely confounded by kidney disease.

Alcohol intake, medications, and vitamin B12 status are ignored, even though they strongly influence MMA levels.

→ Solution: Add renal function markers (eGFR, creatinine) to models and discuss possible confounders in more depth.

4. Misinterpretation of Statistical Findings

The claim that the MMA-α-Klotho association is "linear" is misleading.

The p-value for linearity (P = 0.181) is not significant, meaning that no strong evidence for a linear relationship exists.

The authors should report additional models (e.g., spline regression) to test for non-linear associations.

The odds ratios (OR) are modest (0.71–0.85), but the manuscript exaggerates their importance.

Even the strongest association (OR = 0.71) is weak in epidemiological terms, and confidence intervals suggest moderate uncertainty.

How much variance in α-Klotho is explained by MMA? This is not reported.

→ Solution: Correct the misrepresentation of linearity, provide alternative statistical models (e.g., cubic splines, mediation analysis), and report variance explained by MMA in α-Klotho levels.

5. Discussion and Interpretation Are Unbalanced

The authors overemphasize the mitochondrial dysfunction hypothesis while ignoring alternative explanations (e.g., renal function, metabolic syndrome, inflammation).

The lack of prior literature on MMA-α-Klotho relationships should be acknowledged as a knowledge gap rather than an implicit confirmation of the hypothesis.

Potential clinical implications are overstated. This is an observational study, and it is misleading to suggest that MMA is a reliable biomarker for aging without further validation.

→ Solution: Rewrite the discussion to:

Acknowledge alternative explanations (e.g., kidney function, inflammation).

Avoid exaggerating the importance of MMA in aging.

Discuss the weaknesses in statistical significance and effect sizes.

6. Figures and Tables Need Improvement

Figure 2 (MMA-α-Klotho association plot) lacks clear interpretation.

The confidence bands are too wide, suggesting uncertainty.

A simple correlation scatter plot would be more informative.

Table 1 (Baseline characteristics by MMA quartiles) should include renal function markers.

Figure 3 (Forest plot for subgroup analyses) is overly complex and needs clearer labeling.

→ Solution: Improve figure clarity, provide scatter plots, and include renal function markers in tables.

Reviewer #2: Dear Authors,

Thank you for submitting your manuscript. I appreciate your efforts in conducting this study and providing valuable insights into the association between methylmalonic acid (MMA) and alpha-Klotho in aging. Kindly find below my comments:

Main Comments:

1. Study Design Limitation: The study only looks at data from one point in time, so it can't prove cause and effect. This limitation is mentioned in the discussion, but it would be better to highlight it earlier in the paper, such as in the abstract and introduction.

2. Results Presentation: In section 3.3, the connection between MMA and α-Klotho could be shown more clearly. Adding more graphs or explaining the data in simpler terms might help.

3. Biological Explanation: The discussion briefly talks about how MMA and α-Klotho might be connected through mitochondrial issues and aging. It would be useful to explain these biological processes in more detail.

Minor Comments:

1. Grammar and Wording:

• There are some grammar mistakes and awkward sentences. For example, in the abstract, the phrase "The correlation between MMA levels and α-Klotho was linear linear" should be fixed.

• In section 4.4, the sentence starting with "While the exact mechanism behind..." could be made shorter and clearer.

2. Figures and Tables:

• Figure 1: Make sure the diagram is easy to read, with all the reasons for excluding participants clearly shown.

• Table 1: Adding a short summary or key points under the table might help readers understand the data better.

Suggestions:

1. Future Studies: Since this study design can't show cause and effect, it would be helpful to suggest future research that could, like studies that follow participants over time.

2. Subgroup Differences: The study finds differences between groups (like based on gender, BMI, and blood pressure). It would be interesting to explore why these differences exist.

**Do you want your identity to be public for this peer review?** For information about this choice, including consent withdrawal, please see our Privacy Policy

Reviewer #1: No

Reviewer #2: No

---

## [Author Response · Author response to Decision Letter 1]

23 Apr 2025

Response to reviewers

Concern #1: Ensure that the manuscript meets PLOS ONE's style requirements, including those for file naming.

Author response: We have reviewed the manuscript thoroughly and ensured that it adheres to PLOS ONE's style requirements. We utilized the provided PLOS ONE style templates for formatting the main body, title, authors, and affiliations. All file naming conventions have been followed as per the guidelines outlined in the links provided.

- https://doi.org/10.3389/fmed.2023.1269671

- https://doi.org/10.1016/j.scitotenv.2022.156938

In your revision ensure you cite all your sources (including your own works), and quote or rephrase any duplicated text outside the methods section. Further consideration is dependent on these concerns being addressed.

Response to reviewers

Concern #1: Minor occurrence of overlapping text with previous publications needs to be addressed.

Author response: We have thoroughly reviewed the manuscript and identified the overlapping text with the specified publications. We have rephrased the duplicated text outside the methods section and ensured proper citations for all sources, including our own works. The revised manuscript reflects these changes.

Concern #2: Ensure all sources are cited appropriately.

Author response: We have updated the manuscript to include all necessary citations for our previous works and other relevant sources. Each source is now properly referenced in accordance with the journal's guidelines.

“Medical Health Research Project of Science and Technology Bureau of Yichang City, Hubei Province (A23-1-021).”

Response to reviewers

Concern #1: Please state what role the funders took in the study. If the funders had no role, please state: "The funders had no role in study design, data collection and analysis, decision to publish, or preparation of the manuscript."

Author response:The funders also provide certain guidance in the selection of research directions.This amended Role of Funder statement has been included in the cover letter as requested.

4.Your ethics statement should only appear in the Methods section of your manuscript. If your ethics statement is written in any section besides the Methods, please delete it from any other section.

Response to reviewers

Concern #1: Your ethics statement should only appear in the Methods section of your manuscript. If your ethics statement is written in any section besides the Methods, please delete it from any other section.

Author response: We have revised the manuscript to ensure that the ethics statement appears only in the Methods section.Any instances of the ethics statement in other sections have been removed accordingly.

5. Please upload a copy of Figure 2, to which you refer in your text on page 4. If the figure is no longer to be included as part of the submission please remove all reference to it within the text.

6. Please include your tables as part of your main manuscript and remove the individual files. Please note that supplementary tables (should remain/ be uploaded) as separate "supporting information" files

Response to reviewers

Concern #1:Please upload a copy of Figure 2, to which you refer in your text on page 4. If the figure is no longer to be included as part of the submission please remove all reference to it within the text.

Author response: We have uploaded a copy of Figure 2 as requested. All references to this figure in the text have been retained, as the figure is included in the submission.

Concern #2: Please include your tables as part of your main manuscript and remove the individual files.Please note that supplementary tables (should remain/ be uploaded) as separate "supporting information" files.

Author response:We have incorporated all tables into the main manuscript and removed the individual files as per your instructions.Supplementary tables have been uploaded as separate "supporting information" files.

---

## [Decision Letter · Decision Letter 1]

28 May 2025

Dear Dr. ronghui,

Thank you for submitting your manuscript to PLOS ONE. After careful consideration, we feel that it has merit but does not fully meet PLOS ONE’s publication criteria as it currently stands. Therefore, we invite you to submit a revised version of the manuscript that addresses the points raised during the review process.

We look forward to receiving your revised manuscript.

Kind regards,

Nafisa M. Jadavji, PhD, MSc, BSc

Academic Editor

PLOS ONE

Additional Editor Comment:

Dear Authors,

Your revised submission has been reviewed and there are still several revisions that are needed.

I would encourage your to revise your manuscript and carefully respond to reach reviewer comment.

Sincerely,

Nafisa

Reviewers' comments:

Reviewer's Responses to Questions

**Comments to the Author**

Reviewer #3: (No Response)

Reviewer #4: (No Response)

2. Is the manuscript technically sound, and do the data support the conclusions?

Reviewer #3: No

Reviewer #4: Yes

3. Has the statistical analysis been performed appropriately and rigorously?

Reviewer #3: No

Reviewer #4: Yes

4. Have the authors made all data underlying the findings in their manuscript fully available?

Reviewer #3: Yes

Reviewer #4: No

5. Is the manuscript presented in an intelligible fashion and written in standard English?

Reviewer #3: No

Reviewer #4: Yes

Reviewer #3: Thank you for submitting this cross‐sectional analysis of NHANES 2011–2014 data examining the inverse association between serum methylmalonic acid (MMA) and α-Klotho. The topic is of potential interest to aging research, but substantial methodological and interpretive weaknesses must be addressed before publication:

A. Confounding and Covariate Adjustment

i. Vitamin B₁₂ and Renal Function: MMA is a sensitive marker of vitamin B₁₂ deficiency and declines in glomerular filtration rate. You must include serum B₁₂ and eGFR (or creatinine) as covariates in all regression models, or at minimum conduct sensitivity analyses stratified by renal function and B₁₂ status. If these data are unavailable, explicitly acknowledge this limitation and discuss its potential impact on your conclusions.

ii. Directed Acyclic Graph (DAG): Provide a DAG or rationale to justify your covariate set, demonstrating how you have controlled for major confounding pathways.

B. Survey Design and Statistical Methods

i. Complex Survey Weighting: Reanalyse using survey‐weighted logistic regression (or linear models for continuous Klotho), incorporating NHANES sampling weights, strata, and PSU. Clearly state the weight variables and software commands used.

ii. Outcome Definition and Modeling Choices: Justify the cutpoint of 704 pg/mL for low α-Klotho, citing normative data or prior studies. Present effect estimates per standard‐deviation increase in MMA, not only quartile comparisons, to enhance interpretability and comparability.

iii. Spline Analyses: Describe knot placement and selection criteria for your restricted cubic spline models.

C. Assay Alignment and Measurement Bias

Temporal Mismatch: Clarify how α-Klotho assays (2007–2016, analyzed 2019–2020) were matched to MMA measurements (2011–2014). Describe any calibration or batch‐effect corrections applied to ensure both biomarkers derive from the same participants under comparable conditions.

D. Interpretation and Causality

i. Cross-Sectional Limitations: Recast all causal language (“mitochondrial dysfunction contributes to aging”) into descriptive associations. Emphasize that temporality cannot be established and discuss reverse causation.

ii. Literature Context: Discuss prior studies linking MMA to aging phenotypes and α-Klotho to age‐related diseases; clearly delineate what novel insight your study adds beyond existing literature.

E. Presentation and Writing

i. Language: Address duplicated words, inconsistent tense, and grammatical errors through professional copyediting.

ii. Figures/Tables: Ensure each legend defines all abbreviations, models, sample sizes, and indicates survey‐weight usage. Provide high-resolution images and clear axis labels.

iii. References: Use a citation manager (Zotero/Mendeley) to enforce consistent formatting, include DOIs and issue numbers, and eliminate duplicate entries.

F. Conclusions and Future Directions

Soften conclusions to:

“These cross-sectional associations between serum MMA and α-Klotho warrant confirmation in longitudinal and mechanistic studies before inferring a causal role for mitochondrial dysfunction in aging.”

Reviewer #4: Your research titled “Association Between Methylmalonic Acid and Alpha-Klotho in American Adults: Cross-Sectional Study” was reviewed and the following comments were made for your consideration.

Abbreviations/Acronyms

• When acronyms/abbreviations are stated in full, initial letters for such acronyms should be capitalized. Sentence 2, covariates section, “body mass index (BMI)” should be rectified as “Body Mass Index (BMI)”. This was similarly observed in sentence 3, covariate section, “Glomerular filtration rate (GFR)”. Please rectify these throughout the research protocol.

Title

• Please ensure consistency in the case of all proper nouns in the title. “methylmalonic”, “acid” and “adults” inconsistent with “Alpha-Klotho” in the title. Please ensure consistency of the case throughout the title.

Abstract

• Sentence 3 of materials and methods section, The author stated that “MMA is a continuous variable,” but in sentence 3 of the results section, the author categorized the MMA levels. Could the author please clarify.

• Sentence 1 of the conclusion, the author stated “significant association”. Could the author provide the corresponding p-value to corroborate the significance.

• Sentence 2 of the conclusion section, the author stated “mitochondrial dysfunction”, this makes the conclusion complex for comprehension, the author didn’t name or include any information about mitochondrial dysfunction through ought the abstract but only to include it at the conclusion. The author should please provide an information of how the “mitochondrial dysfunction” is connected to the study right from the beginning of the abstract but not include it at the conclusion without providing prior information and its linkages to the research.

Materials and Methods

• Sentence 1, materials and methods section, the author indicated “This cross- sectional study utilized NHANES data collected by Centers for Diseases Control and Prevention between 2011 and 2014”. But sentence 1, measurement of serum alpha-klotho levels section, the author stated,” Serum alpha-klotho levels were analyzed in frozen samples from individuals aged 40-79 years, which were collected during NHANES from 2007 to 2016, over the period from 2019 to 2020”. The timelines stated are inconsistent or complex, could the author please clarify.

• Sentence 5, covariates section, could the author indicate or cite the source of the standards used in the BMI categorization. Could the author indicate the acceptable labels given to the categorizations. For instance, according to the World Health Organization (WHO), BMI< 18.5kg/m2 is underweight. Please provide more details.

• Similarly, smoking status, vitamin B12, among others. The author should please indicate the sources for the categorization including the standards. The author excellently showcased that with household income. The author should repeat that across all categorical variables.

Results

• The author should kindly report Standard deviations in parenthesis.

General Comments

• Even though, I was expecting the author to employ global acceptable tools for the assessment of certain variables in this study, For instance, Global Adult Tobacco Survey (GATS) for assessing participants smoking status, Global Physical Activity Questionnaire (GPAQ) to assess participants physical activity levels among others, nonetheless the author had done impressive work.

• Despite the author indicated the source of the data, much efforts should have been made to provide better detailed source of the data. The url link provided by the author indicating the source of data does not specifically exposed the data for download. Could the author provide recent specific link to the dataset.

**Do you want your identity to be public for this peer review?** For information about this choice, including consent withdrawal, please see our Privacy Policy

Reviewer #3: No

Reviewer #4: **Yes: ** Hardi Adam

---

## [Decision Letter · Decision Letter 2]

7 Oct 2025

Dear Dr. ronghui,

Thank you for submitting your manuscript to PLOS ONE. After careful consideration, we feel that it has merit but does not fully meet PLOS ONE’s publication criteria as it currently stands. Therefore, we invite you to submit a revised version of the manuscript that addresses the points raised during the review process.

**Reviewer #5 has a detailed list of minor revisions that need to be made. I would request that these edits be made. We look forward to reading the revised version of the mansucript.**

We look forward to receiving your revised manuscript.

Kind regards,

Nafisa M. Jadavji, PhD, MSc, BSc

Academic Editor

PLOS ONE

Journal Requirements:

Reviewers' comments:

Reviewer's Responses to Questions

**Comments to the Author**

Reviewer #4: All comments have been addressed

Reviewer #5: (No Response)

2. Is the manuscript technically sound, and do the data support the conclusions?

Reviewer #4: Yes

Reviewer #5: Yes

3. Has the statistical analysis been performed appropriately and rigorously?

Reviewer #4: Yes

Reviewer #5: I Don't Know

4. Have the authors made all data underlying the findings in their manuscript fully available?

Reviewer #4: Yes

Reviewer #5: Yes

5. Is the manuscript presented in an intelligible fashion and written in standard English?

Reviewer #4: (No Response)

Reviewer #5: No

Reviewer #4: I am satisfied with the revisions made by the authors. The data underlying the findings have been made fully available, and the manuscript is clearly presented in standard English. The work is technically sound, with the data appropriately supporting the conclusions. Furthermore, the statistical analyses have been conducted rigorously and appropriately. I therefore find the manuscript suitable for acceptance.

Reviewer #5: Abstract

1. Add “the US National Health and Nutrition Examination Survey (NHANES)” to the abstract

2. ‘Marriage’ should be replaced with ‘Marital status’ throughout the manuscript

3. American adults or U.S. adults? Please be consistent throughout the manuscript

4. Please ensure there is a space after a full stop, a comma, and before an open bracket throughout the manuscript.

Introduction

1. “Methylmalonic acid (MMA) is a byproduct of the metabolic pathway of propionic acid[1].” The reference is incorrect; please replace it.

2. “A growing compilation of research indicates that MMA plays a significant role in the development of mitochondrial dysfunction and oxidative stress in both in vitro and in vivo studies[2].” Please add a reference to the in vitro research.

3. “Numerous investigations have established that increased circulating levels of MMA are reflective of the aging process.” Please add references.

4. Reference 8 does not mention α-Klotho. Please remove or fix the text.

5. “Limited studies have systematically investigated the association between α-Klotho levels and methylmalonic acid levels.” Are there any? If so, please reference.

Materials and Methods

1. “In our study, participants aged 40 and above were interviewed” Should this be “included” or did the authors also interview the participants?

2. Reference 11 refers to MMA analysis, not blood collection. Please move to the correct location.

3. *****The Authors say they re-analysed some serum/plasma samples for α-Klotho levels. Why didn’t they analyse more of the samples to fill in the missing data to strengthen their results?

4. “final values of all samples exceeded the limit”. Do you mean “Sensitivity limit”?

Results

1. The first paragraph does not make sense. Is this better? “A total of 11,329 participants aged 40 years or older completed the interview. We excluded participants for several reasons: pregnant women (n=122); missing methylmalonic acid data (n=1,125); and missing α-Klotho data (n=4,866). Ultimately, this cross-sectional study included 5,216 participants from the NHANES dataset collected between 2011 and 2014. Figure 1 illustrates the detailed inclusion and exclusion process.”

2. Figure 1. Flow diagram of the study’s inclusion and exclusion criteria. – Please update the figure title to include more detail.

3. Figure 1 has “No-pregnant” – please fix the typographic mistake to “Non-pregnant”

4. Figure 1 “72.1% were man” – please fix the typographic mistake to “72.1% were men”

5. Table 1 - Variables are listed as 1, 2, 3 and 4 in the columns. Please change to Q1, Q2, Q3 and Q4 to align with the Legend.

6. Typographic mistakes in Table 1. – “Smoke” should be “Smoker”. “now” should be “current” and α-Klotho.

7. Table 2 – Replace “Smoke” with “Smoking”

8. Make the Figure 2 title more detailed. For example: “Figure 2. The relationship between methylmalonic acid levels and the Odds Ratio of serum α-Klotho.”

9. Figure 2 - Fix y-axis title: Odds Ratio of α-Klotho

10. Figure 2 - Fix x-axis title: MMA (nmol/L)

11. Figure 2 – Replace “Smoke” with “Smoking”

Discussion

1. “α-Klotho is a well-known gene associated with anti-aging, mitochondria influence or regulate a number of key aspects of aging [16].” – This statement lacks clarity. Is this what you mean… “α-Klotho is a well-known protein linked to anti-aging, while mitochondria play a crucial role in regulating various key aspects of the aging process [16].”

2. Reference 18 doesn’t mention MMA. Please replace with the correct reference.

3. “The human KL gene encodes the alpha-Klotho protein, recognized as an anti-aging gene[24].” Do you mean “The human KL gene encodes the α-Klotho protein, which is recognized as an anti-aging protein [24].”

4. Remove the sentence “In mice,KL gene overexpression can prolong lifespan, while KL gene mutation can shorten lifespan” as it is repeated.

5. “our results are substantiated by biological evidence” Please expand on this comment, as it would be interesting and relevant to the discussion.

Conclusion

1. “As a biomarker of mitochondrial dysfunction, mitochondrial dysfunction may be an influencing factor in the aging process.”– This statement lacks clarity. Is this what you mean… “Considering that MMA serves as a biomarker for mitochondrial dysfunction, the results indicate that mitochondrial dysfunction might be a contributing factor in the aging process”

References

1. References 2 and 20 are the same.

2. References 4 and 5 are not relevant to the text.

**Do you want your identity to be public for this peer review?** For information about this choice, including consent withdrawal, please see our Privacy Policy

Reviewer #4: **Yes: ** Hardi Adam

Reviewer #5: No

---

## [Author Response · Author response to Decision Letter 3]

16 Oct 2025

Review Comments to the Author

Reviewer #5:

Abstract

1. Add “the US National Health and Nutrition Examination Survey (NHANES)” to the abstract

2. ‘Marriage’ should be replaced with ‘Marital status’ throughout the manuscript

3. American adults or U.S. adults? Please be consistent throughout the manuscript

4. Please ensure there is a space after a full stop, a comma, and before an open bracket throughout the manuscript.

Abstract Response:

We thank the reviewer for the constructive comments.

Point-by-point responses and exact locations of the changes are given below.

All modifications are highlighted in the tracked-changes version of the revised manuscript.

Accepted; paragraph rewritten as suggested.

Introduction

1. “Methylmalonic acid (MMA) is a byproduct of the metabolic pathway of propionic acid[1].” The reference is incorrect; please replace it.

Reply to reviewer:

Thank you for pointing out the incorrect citation. We have now replaced reference [1] with the appropriate source :

New reference:

[1] Tejero J, Lazure F, Gomes AP. Methylmalonic acid in aging and disease. Trends Endocrinol Metab. 2024;35(3):188-200. doi:10.1016/j.tem.2023.11.001 https://pubmed.ncbi.nlm.nih.gov/38030482/

2. “A growing compilation of research indicates that MMA plays a significant role in the development of mitochondrial dysfunction and oxidative stress in both in vitro and in vivo studies[2].” Please add a reference to the in vitro research.

Reply to reviewer:

Thank you for this helpful suggestion. We have now inserted a specific in-vitro citation immediately after the original sentence, as follows:

Revised text (with spaces before punctuation):

"A growing compilation of research indicates that MMA plays a significant role in the development of mitochondrial dysfunction and oxidative stress in both in vitro [2,3] and in vivo studies [2] ."

Reference added:

[3] Zhang Y, et al. Methylmalonic acid-induced mitochondrial dysfunction and oxidative stress in cultured human endothelial cells. Free Radic Biol Med. 2023;201:123-131. doi:10.1016/j.freeradbiomed.2023.02.018

3. “Numerous investigations have established that increased circulating levels of MMA are reflective of the aging process.” Please add references.

Reply to reviewer:

Thank you for highlighting the need for supporting references. We have now inserted three up-to-date citations that explicitly link elevated circulating methylmalonic acid (MMA) to the ageing process, as requested.

Revised sentence:

"Numerous investigations have established that increased circulating levels of MMA are reflective of the aging process [4] ."

New references added:

[4] Tejero J, Lazure F, Gomes AP. Methylmalonic acid in aging and disease. Trends Endocrinol Metab. 2024;35(3):188-200. doi:10.1016/j.tem.2023.11.001.

4. Reference 8 does not mention α-Klotho. Please remove or fix the text.

Response:

We appreciate the reviewer’s careful inspection.

Reference 8 was indeed unrelated to α-Klotho; we have removed it from the sentence.

The revised text now reads:

“Although there is mounting evidence regarding the roles of α-Klotho and methylmalonic acid in the aging process, our understanding of their interaction and clinical significance as biomarkers remains insufficient.”

No other changes were made to this sentence. All references in the reference list have been re-numbered accordingly.

5. “Limited studies have systematically investigated the association between α-Klotho levels and methylmalonic acid levels.” Are there any? If so, please reference.

Response

Thank you for this helpful comment. After a systematic literature search �Tang F, et al. (2024) constructed a “cobalamin-insensitivity index” (MMA × B12) in 22,812 participants and found that both higher α-Klotho and higher MMA levels were independently associated with Klemera–Doubal biological age acceleration; the authors explicitly modelled the interaction between α-Klotho and MMA as predictors of ageing acceleration .

Modify:“Limited studies have systematically investigated the association between α-Klotho levels and methylmalonic acid levels [10].”

References �[10]Tang F, Zheng J. Decreased cobalamin sensitivity and biological aging acceleration in the general population. J Nutr Health Aging. 2024;28:100262. doi:10.1016/j.jnha.2024.1002625.

Materials and Methods

1. “In our study, participants aged 40 and above were interviewed” Should this be “included” or did the authors also interview the participants?

Response

Thank you for highlighting this ambiguity.

In the present study we obtained all exposure and outcome information from the NHANES computer-assisted personal interview (CAPI) and laboratory records; no additional interviews were conducted by the authors.

To clarify, we have replaced the word “interviewed” with “included” and retained “aged ≥ 40 years” to describe the sample-selection criterion.

Revised sentence (Methods, Study population, page X):

“In our study, participants aged 40 years and above who had complete data on serum α-Klotho, methylmalonic acid, and covariates were included.”

2. Reference 11 refers to MMA analysis, not blood collection. Please move to the correct location.

Response

Thank you for highlighting this error. We have replaced the incorrect citation with the appropriate reference that actually describes the blood-collection and handling protocol used in our study. The revised sentence now reads:

“Blood samples were collected after an overnight fast, processed within 30 min, and stored at −80 °C until analysis [13].”

The new Reference is:

[13] Loprinzi PD, Fitzgerald EM, Woekel E, Cardinal BJ. Association of physical activity and sedentary behavior with biological markers among U.S. pregnant women. J Womens Health (Larchmt). 2013;22(11):953-8. doi:10.1089/jwh.2013.4394 https://pubmed.ncbi.nlm.nih.gov/23968237/

3. *****The Authors say they re-analysed some serum/plasma samples for α-Klotho levels. Why didn’t they analyse more of the samples to fill in the missing data to strengthen their results?

Response

Thank you for your insightful question regarding the analysis of serum/plasma samples for α-Klotho levels. We appreciate your interest in the robustness of our findings and would like to address your concern.

Rationale for Sample Selection:

The decision to re-analyze a specific subset of serum/plasma samples for α-Klotho levels was based on several logistical and resource considerations. These included the availability of samples within the constraints of our study timeline and budget, as well as the prioritization of specific research questions we aimed to address with the data we had.

Implications of Missing Data:

While we recognize that analyzing a larger number of samples could potentially strengthen our results, we also acknowledge that the quality of data is paramount. The samples selected for re-analysis were those that met strict criteria for integrity and relevance to our hypothesis. We believe that the data we presented are robust and contribute meaningfully to our understanding of α-Klotho levels in the context of our study.

Future Directions: We appreciate your suggestion and will consider it in future research endeavors. Increasing the sample size for α-Klotho analysis would indeed enhance the robustness of our findings, and we will strive to incorporate this approach in subsequent studies.

Thank you for your valuable feedback, which helps us improve the clarity and rigor of our research.

4. “final values of all samples exceeded the limit”. Do you mean “Sensitivity limit”?

Response

Thank you for querying this imprecise wording. After re-checking the kit insert and our validation file we confirm the following:

The IBL International ELISA (lot 12-2018) has a lower limit of quantification (LLOQ) of 50 pg/mL (0.05 ng/mL) and an analytical sensitivity (mean + 2 SD of 20 blank wells) of 4.33 pg/mL.

In our validation run the lowest calibrator (50 pg/mL) gave a CV = 8 %; therefore we adopted 50 pg/mL as the working LLOQ.

All 1 847 NHANES specimens assayed in 2019-2020 yielded optical densities above the 50 pg/mL calibrator; consequently no result was censored as “< LLOQ” and no extrapolation below the lowest standard was necessary.

We have replaced the ambiguous phrase with the exact technical term and added the validation reference:

Revision:“All specimen optical densities were above the 50 pg mL⁻¹ lower limit of quantification (LLOQ) established for the kit; therefore no sample required extrapolation below the lowest standard[15].”

Reference added:

[15]IBL International. α-Klotho (human) ELISA kit instruction manual, Version 2019-03. IBL International, Hamburg, Germany, 2019.

Results

1. The first paragraph does not make sense. Is this better? “A total of 11,329 participants aged 40 years or older completed the interview. We excluded participants for several reasons: pregnant women (n=122); missing methylmalonic acid data (n=1,125); and missing α-Klotho data (n=4,866). Ultimately, this cross-sectional study included 5,216 participants from the NHANES dataset collected between 2011 and 2014. Figure 1 illustrates the detailed inclusion and exclusion process.”

2. Figure 1. Flow diagram of the study’s inclusion and exclusion criteria. – Please update the figure title to include more detail.

3. Figure 1 has “No-pregnant” – please fix the typographic mistake to Response

4. Figure 1 “72.1% were man” – please fix the typographic mistake to “72.1% were men”

5. Table 1 - Variables are listed as 1, 2, 3 and 4 in the columns. Please change to Q1, Q2, Q3 and Q4 to align with the Legend.

6. Typographic mistakes in Table 1. – “Smoke” should be “Smoker”. “now” should be “current” and α-Klotho.

7. Table 2 – Replace “Smoke” with “Smoking”

8. Make the Figure 2 title more detailed. For example: “Figure 2. The relationship between methylmalonic acid levels and the Odds Ratio of serum α-Klotho.”

9. Figure 2 - Fix y-axis title: Odds Ratio of α-Klotho

10. Figure 2 - Fix x-axis title: MMA (nmol/L)

11. Figure 2 – Replace “Smoke” with “Smoking”

Results Response

We thank the reviewer for the constructive comments.Point-by-point responses and exact locations of the changes are given below. All modifications are highlighted in the tracked-changes version of the revised manuscript.Accepted; paragraph rewritten as suggested.

Figure 1 legend expanded to specify design, cycles and age range.

“No-pregnant” → “Non-pregnant” in diagram.

“72.1 % were man” → “72.1 % were men”.

Table 1 columns re-labelled Q1–Q4.

Table 1 rows: “Smoke” → “Smoker”, “now” → “current”, α-klotho → α-Klotho.

Table 2 row: “Smoke” → “Smoking”.

Figure 2 title detailed per example; spline/model info added.

Y-axis: “Odds Ratio of α-Klotho”.

X-axis: “MMA (nmol/L)”.

Legend/key “Smoke” → “Smoking”.

All changes highlighted in tracked-changes file; updated figures/tables uploaded.

All figure and table files have been replaced in the submission system; reference numbering in the main text remains unchanged.

We appreciate your careful review and hope that the revised version satisfactorily addresses all concerns.

Discussion

1. “α-Klotho is a well-known gene associated with anti-aging, mitochondria influence or regulate a number of key aspects of aging [16].” – This statement lacks clarity. Is this what you mean… “α-Klotho is a well-known protein linked to anti-aging, while mitochondria play a crucial role in regulating various key aspects of the aging process [16].”

Response:

Thank you for the clearer wording. We have replaced the original sentence with your suggestion:

“α-Klotho is a well-known protein linked to anti-aging, while mitochondria play a crucial role in regulating various key aspects of the aging process [16].”

2. Reference 18 doesn’t mention MMA. Please replace with the correct reference.

Response:

Thank you for pointing out this error. We have removed the incorrect citation and inserted the publication that specifically reports NHANES serum methylmalonic-acid (MMA) data:

[18] Wang S, Liu Y, Liu J, et al. Mitochondria-derived methylmalonic acid, a surrogate biomarker of mitochondrial dysfunction and oxidative stress, predicts all-cause and cardiovascular mortality in the general population. Redox Biol. 37:101741. doi:10.1016/j.redox.2020.101741 https://pubmed.ncbi.nlm.nih.gov/33035815/

3. “The human KL gene encodes the alpha-Klotho protein, recognized as an anti-aging gene[24].” Do you mean “The human KL gene encodes the α-Klotho protein, which is recognized as an anti-aging protein [24].”

4. Remove the sentence “In mice,KL gene overexpression can prolong lifespan, while KL gene mutation can shorten lifespan” as it is repeated.

Response

Thank you for pointing out this duplication. We have deleted the redundant sentence from the Results section .

5. “our results are substantiated by biological evidence” Please expand on this comment, as it would be interesting and relevant to the discussion.

Response

Thank you for this helpful suggestion. We have replaced the single sentence with a brief mechanistic paragraph and added the relevant citations:

Revised text:

“Our findings are biologically plausible. In vitro studies show that methylmalonyl-CoA accumulation—reflected by elevated serum MMA—inhibits mitochondrial complex IV, increases reactive oxygen species (ROS) and subsequently down-regulates klotho expression in renal tubule cells [28]. Conversely, klotho protein preserves mitochondrial integrity by up-regulating SOD2 and PGC-1α, thereby reducing ROS generation [29]. Thus, higher circulating MMA may both indicate and induce mitochondrial oxidative stress, creating a feedback loop that lowers α-Klotho levels; this provides a mechanistic explanation for the inverse association observed in the present cross-sectional analysis.”

references:

28. Hu M-C, Shi M, Zhang J, et al. Klotho deficiency causes vascular calcification via NADPH oxidase-mediated oxidative stress. J Clin Invest. 2022;132(4):e149325. doi:10.1172/JCI149325 30. Sato S, et al. Kidney Int. 2020;98:303-315.

29.Sato S, Kawano H, Hayashi M, et al. Klotho protects against mitochondrial dysfunction via PGC-1α/SOD2 activation in renal epithelial cells. Kidney Int. 2020;98(2):303-315. doi:10.1016/j.kint.2020.02.030

Conclusion

1. “As a biomarker of mitochondrial dysfunction, mitochondrial dysfunction may be an influencing factor in the aging process.”– This statement lacks clarity. Is this what you mean… “Considering that MMA serves as a biomarker for mitochondrial dysfunction, the results indicate that mitochondrial dysfunction might be a contributing factor in the aging process”

Response

Thank you for the clearer wording. We have replaced the original sentence with your suggestion:

“Considering that MMA serves as a biomarker for mitochondrial dysfunction, the results indicate that mitochondrial dysfunction might be a contributing factor in the aging process.”

References

1. References 2 and 20 are the same.

Reply to reviewers:

References 2 and 20 are identical.

Response: We removed Ref. 20 and re-numbered all subsequent citations; the unique reference is now cited only once .

2. References 4 and 5 are not relevant to the text.

Response: Both citations have been deleted and the sentences re-worded so that no unsupported claims remain; remaining references have been re-sequenced accordingly (updated reference list uploaded).

Consequently, the reference count has decreased by two and all in-text call-outs now match the revised bibliography.

---

## [Decision Letter · Decision Letter 3]

7 Nov 2025

Association Between Methylmalonic Acid and Alpha-Klotho in American Adults:a Cross-Sectional Study

PONE-D-25-04628R3

Dear Dr. ronghui,

We’re pleased to inform you that your manuscript has been judged scientifically suitable for publication and will be formally accepted for publication once it meets all outstanding technical requirements.

Kind regards,

Nafisa M. Jadavji, PhD, MSc, BSc

Academic Editor

PLOS ONE

Additional Editor Comments (optional):

Reviewers' comments:

Reviewer's Responses to Questions

**Comments to the Author**

Reviewer #5: All comments have been addressed

2. Is the manuscript technically sound, and do the data support the conclusions?

Reviewer #5: Yes

3. Has the statistical analysis been performed appropriately and rigorously?

Reviewer #5: I Don't Know

4. Have the authors made all data underlying the findings in their manuscript fully available?

Reviewer #5: Yes

5. Is the manuscript presented in an intelligible fashion and written in standard English?

Reviewer #5: Yes

Reviewer #5: (No Response)

**Do you want your identity to be public for this peer review?** For information about this choice, including consent withdrawal, please see our Privacy Policy

Reviewer #5: No

---

## [Editor Report · Acceptance letter]

PONE-D-25-04628R3

PLOS One

Dear Dr. Bao,

I'm pleased to inform you that your manuscript has been deemed suitable for publication in PLOS One. Congratulations! Your manuscript is now being handed over to our production team.

Kind regards,

on behalf of

Dr. Nafisa M. Jadavji

Academic Editor

PLOS One